# An Analysis of Veterinary Practitioners’ Intention to Intervene in Animal Abuse Cases in South Korea

**DOI:** 10.3390/ani10050802

**Published:** 2020-05-06

**Authors:** Seola Joo, Yechan Jung, Myung-Sun Chun

**Affiliations:** 1Research Institute for Veterinary Science, College of Veterinary Medicine, Seoul National University, Seoul 08826, Korea; jsa321@snu.ac.kr; 2Research Ethics Center, Korea University, Seoul 02841, Korea; vethics@korea.ac.kr

**Keywords:** animal abuse, veterinary practitioner, reporting, education, forensic science, human crimes, animal safety, abuse prevention, interventions

## Abstract

**Simple Summary:**

In South Korea, from 2013–2018, the number of animal abuse crimes has increased by 3.3 times. Since veterinarians are at the forefront of witnessing and identifying such issues, we aimed to investigate the perceived barriers and factors that influence their intention toward reporting animal abuse cases and counseling animal owners in South Korea to develop insights into how they can be encouraged to engage in abuse prevention. We collected data regarding the number of cases witnessed, responses to these cases, barriers associated with reporting cases, belief in the link between animal abuse and human crimes and moral and legal obligations, and participant characteristics via a questionnaire. On analyzing the responses, we uncovered that 80% of the participants witnessed suspected animal abuse cases, and more than half of them were reluctant to report them to the authorities. We found that a “pro-animal” attitude and belief in the “link” between animal abuse and human abuse and moral and legal obligations toward animals are significantly associated with intervening in animal abuse cases. To reinforce these aspects, we recommend that veterinarians be trained in legal liability, moral responsibility, and veterinary forensic medicine. To overcome perceived barriers, legislation to protect victims and reporting veterinarians should be introduced.

**Abstract:**

Due to their professional abilities, veterinarians have a duty to reduce animal abuse. Therefore, it is crucial to understand their attitude and behavior on encountering animal abuse cases. We analyzed the responses from completed questionnaires (*n* = 593) filled by small and large animal practitioners in South Korea. The majority (*n* = 513, 86.5%) of the respondents witnessed suspected animal abuse cases in their practice. The female participants, small animal practitioners, and younger veterinarians tended to report animal abuse cases more frequently. Based on a hierarchical regression model, moral obligation was the statistically significant predictor of intention toward counseling owners (F = 22.089, R^2^ = 0.232, *p* < 0.001) while “pro-animal” attitudes, belief in the “link” between animal and human crimes, and moral and legal obligation were significant predictors of intention to report (F = 22.877, R^2^ = 0.239, *p* < 0.001). The most frequent barrier in reporting abuse cases was the difficulty in assuring animal safety afterwards. Our findings revealed that individual characteristics (sex, age, practice type, pro-animal attitude) affect veterinarian sensitivity in recognizing animal abuse. Participants lacked self-efficiency in managing animal abuse cases. Therefore, strengthening professionalism and training veterinarians in identifying nonaccidental injuries caused by abuse are recommended to motivate them to intervene in abuse cases.

## 1. Introduction

Animal abuse is not only an animal welfare issue, but it is also a critical concern to the wellbeing of humans. It is considered as a sign of antisocial behavior trends [1] based on the potential “link” between animal abuse and child abuse or other forms of domestic violence [2,3,4,5,6]. Therefore, every member of a society shares the same responsibility with respect to reducing animal cruelty; however, veterinarians should have a stronger sense of duty in this regard compared to any other group because of their professional abilities [7,8,9]. Therefore, in many countries, animal welfare legislation and the professional policies for veterinarians clarify their obligation to report animal cruelty. For example, in the UK, if an inspection of an animal leads to a diagnosis of nonaccidental injury (NAI) indicating a suspicion of abuse, the veterinarian should first attempt to discuss these concerns with the client. If the veterinarian deems that this situation would be unmanageable, the case should be reported to the relevant authorities. Breaching the obligations associated with client confidentiality is justified in case of serious situations [10]. Similarly, the Canadian Veterinary Medical Association (CVMA) has stated that “society has an obligation to support those veterinarians who report in good faith using their professional judgment” [11]. The provincial government of Quebec, Canada, has stronger legislation outlining that a veterinary surgeon should report their observations to relevant authorities without delay, and if the allegations were found to be accurate, the responsible party would be guilty of an offense and liable to pay a fine [12]. Currently, in the United States, approximately 17 states have legislated that it is mandatory for veterinarians to report any suspected animal abuse cases [13,14]. However, in many other countries, reporting animal abuse is still left up to the discretion of the individual professional [8].

Physicians that encounter cases involving “battered child syndrome,” a clinical condition in young children who have been subjected to severe physical abuse, can play an essential role in initiating a proper investigation in order to ensure the adequate management of the case. It is the physician’s foremost duty and responsibility toward the child to completely evaluate the issue and to ensure that a repetition of the incident that caused suffering to the child would never occur [15]. Likewise, veterinarians are at the forefront of witnessing and identifying abused animals or “battered pets” [16]. In previous studies that have focused on veterinarian intervention in animal abuse cases, researchers have highlighted the distinctive role that veterinarians could play in reducing the incidence of animal abuse by identifying cases involving NAIs, and they have also highlighted a possible link between animal abuse and the early detection of child and woman abuse [5,7,17,18,19,20]. Consequently, they explored the factors that could influence a veterinarian’s intention to intervene, such as personal characteristics, experience levels, and perceived barriers associated with reporting animal abuse cases. Cross-sectional studies have investigated veterinarian-related factors, such as experience level, attitude, or knowledge regarding animal abuse or the adequate response to such a situation. The results of these studies indicate that perceptional and behavioral differences are associated with several variables including sex, age, type of practice, belief in their obligation, knowledge regarding the aforementioned “link,” education or training level, culture, and region [20,21,22,23,24]. Female veterinarians were found to have a higher level of agreement and support toward reporting animal abuse cases [21] and advocating mandatory reporting [22,24]. Furthermore, younger (<40 years old) veterinarians had stronger intentions in favor of intervening in animal abuse cases [23]. Compared to large or mixed animal practitioners, small animal practitioners were identified to advocate more strongly for mandatory reporting [24]. Veterinarians were reluctant to report animal abuse cases because of the fear of compromising the safety of the victims and distrust toward the capability of relevant authorities in effectively handling the cases [22,25,26]. They were also concerned about breaching client –veterinarian confidentiality [21], losing income, harming their reputation [21,22,25], and retaliation or legal repercussions [22,25]. They were not sure about how to handle animal abuse cases due to a lack of training, knowledge, and available resources [22,25,27]. Therefore, previous studies have emphasized making animal abuse reporting mandatory by law and on ensuring legal immunity to support the veterinarians who report on such cases [7,28,29].

Recently, the Korean Supreme Prosecutors’ Office reported that over the last five years, from 2013 to 2018, the number of animal abuse crimes has increased by 3.3 times [30]. The increasing companion animal population and rising awareness with respect to animal welfare in South Korea have made it necessary for veterinarians to play an active role against animal abuse cases. The Korean Animal Protection Act (KAPA) has obligated veterinarians to report on animal abuse to the relevant authorities without the fear of incurring a penalty. Therefore, the appropriate attitude and intention with respect to intervening in animal abuse cases are essential in order to reduce their prevalence. For this reason, in this study, we aimed to investigate the perceived barriers and factors that influence a veterinarian’s intention to report animal abuse cases and to counsel the animal owners in South Korea in order to develop insights into how veterinarians can be encouraged to actively engage in animal abuse prevention.

## 2. Materials and Methods

### 2.1. Questionnaire

Our questionnaire was designed to explore the predictive variables (e.g., demographic characteristics, beliefs, attitudes) and their correlations with the intention of South Korean veterinary practitioners toward reporting animal abuse cases and counseling the individuals involved and the barriers associated with appropriately responding to suspected animal abuse cases (Appendix A. Questionnaires). 

The first section of the questionnaire included questions focused on identifying whether respondents had witnessed animal abuse cases involving “deliberate, physical maltreatment or neglect resulting in symptoms requiring veterinary treatment” [21], and, in the case that they had, there was a question on how often they encountered such cases in their practice. An open question was included for the respondents to describe the animal abuse cases that they experienced in the past three years. In the second section, the questions were focused on the veterinary practitioners’ intervention in the case of animal abuse in terms of “counseling owners” and “reporting to relevant authorities,” and the responses were scored on a 5-point Likert scale (1: “definitely do not” to 5: “definitely do”). A checklist of barriers that could lead to veterinary practitioners being reluctant to take an action in suspected animal abuse cases was also provided in this section. The list of choices in which overlapping selections were allowed is as follows: the difficulty in assuring the animal’s safety and welfare after reporting suspected animal abuse, uncertainty in identifying animal abuse, lack of knowledge about how to stop abuse and support victims, fear of losing income and clients, and breaching of clients’ confidentiality. The subsequent section investigated the participants’ belief in the “link” between animal abuse and domestic violence in humans (belief—abuse) or other types of crimes (belief—crime) and the veterinary practitioner’s attitude toward a moral and legal obligation to intervene in animal abuse cases. The level of agreement to the statements corresponding to each question was scored on a 5-point Likert scale (1: “strongly disagree” to 5: “strongly agree”). The last section collected the demographic data, including the sex, age, and type of practice, of the participants. The “pro-animal” attitude was measured using the animal attitude scale (AAS) that consists of ten items and that is one of the most widely used approaches to investigate the ethical and behavioral aspects of human–animal interactions [31].

### 2.2. Data Collection and Statistical Analysis

After receiving ethical approval from the Institutional Review Board of Seoul National University (IRB No. E 1811/003-001), the animal abuse questionnaires (22 items) were presented to the veterinarians (*n* = 6800) via an online form (Survey Monkey) as a part of the questionnaires on veterinary ethics with the help of the Korean Veterinary Medical Association (KVMA) from 20 to 30 November 2018. Participants provided informed consent before partaking in the survey. We received a total of 1818 responses (26.7% response rate) to the survey on ethical issues in Korea, in which a section of animal abuse questions exclusively for small and large animal practitioners were included. In this survey we selected the 593 responses from small and large animal practitioners (32.6% of the total responses) and analyzed the animal abuse questions. We hypothesized that there would be a significant correlation among individual characteristics (age, sex, practice type, and pro-animal attitude), belief in the “link” and in a sense of duty, and the intention of intervening in animal abuse cases. We conducted a descriptive analysis to compute the data and a t-test and analysis of variance (ANOVA) to compare the findings. Furthermore, a hierarchical multiple regression analysis was performed to identify the predictors of the behavioral intention toward counseling owners and reporting in animal abuse cases. We used the Statistical Package for the Social Sciences (SPSS) 23.0 for the statistical analysis.

## 3. Results

### 3.1. Respondents’ Demographic Characteristics and Responses to Witnessing Animal Abuse

The chosen questionnaire responders (*n* = 593, 76.9% male and 23.1% female) represented approximately 8.5% of the total veterinary practitioners in Korea (*n* = 6972 in January of 2020 per the KVMA statistics). The majority of the responders were in their 40’s (37.4%), followed by in their 30’s (29.2%). Approximately 86% of the participants were small animal practitioners, as shown in Table 1. Participants represented the population of veterinary practitioners in Korea, in terms of gender (male 79.5%, female 20.5%), practice type (small animal 86.9%, large animal 13.1%), and age (under the 50’s 68.8%).

The female participants (t = −10.08, *p* < 0.001, Cohen’s d = 0.94) and small animal practitioners (t = −4.92, *p* < 0.001, Cohen’s d = 0.61) exhibited a stronger “pro-animal” attitude. Regarding the age groups, we found a statistically significant difference with a moderate effect size corresponding to a “pro-animal” attitude (F = 6.675, *p* < 0.001, Eta-square = 0.054). A post hoc comparison using Tukey’s honestly significant difference (HSD) test revealed that the participants who were below 50 years old had a stronger “pro-animal” attitude than their counterparts from older age groups.

The majority (*n* = 513, 86.5%) of the responding veterinarians had witnessed suspected animal abuse cases in their practice. The results of a crosstab analysis of each subgroup (sex, age, and practice type) implied that the respondents in the groups comprised of female participants (χ ^2^ = 7.313, *p* < 0.01), small animal practitioners (χ^2^ = 20.937, *p* < 0.001), and younger veterinarians (χ^2^ = 23.537, *p* < 0.001) tended to respond that they had witnessed animal abuse cases.

More than half of the participants (59.6%) had witnessed suspected animal abuse cases one to three times per year, as shown in Table 2. Moreover, 11.1% of all the participants came across abuse cases more than once a month. Furthermore, several participants answered the open question regarding the types of witnessed animal abuse cases (*n* = 169). Per the classification based on the typology of companion animal abuse [32], the most frequent type of abuse among the cases was categorized as passive physical or active mental neglect (*n* = 101), which included the refusal of adequate veterinary medical treatment, abandonment, and the lack of food or care. The other cases (*n* = 68) involved active physical neglect with serious violence or direct harm to animals including fractures induced by kicking and injuries induced by falling, burning, or performing home surgery.

### 3.2. Belief in the “Link” between Animal Abuse and Violence toward Humans and Belief in a Veterinarian’s Obligation

Over 70% of the respondents agreed to the correlation of animal abuse with domestic abuse (strongly agree—33.2%, agree—40.5%) and other crimes (strongly agree—34.2%, agree—41.3%). The participants agreed with their moral obligation (strongly agree—16.2%, agree—55.6%) and legal obligation (strongly agree—6.1%, agree—22.8%) differently. Only 1.9% of the respondents believed that they do not have a moral obligation; meanwhile, 13.7% of the participants believed that they do not have any legal responsibility to intervene in suspected animal abuse cases (Table 3).

### 3.3. Interventions and Barriers Related to Intervening in Animal Abuse Cases

Several participants exhibited reluctance to report animal abuse cases to the authorities as displayed in Table 4. Three-quarters of the participants responded that they would counsel owners in suspected abuse situations (“definitely do”—35.1%, “probably do”—39.5%) while half of them responded that they would report suspected abuse cases (“definitely do”—15%, “probably do”—33.6%).

The most significant barrier associated with intervening in suspected animal abuse cases was found to be the perceived difficulty in assuring the animal’s safety and welfare after reporting a suspected animal abuse case (Table 5). Approximately 70% of the participating veterinarians considered that the current legal and police systems are not capable of securing the safety and welfare of victims. The rest of the selected barriers perceived in reporting abuse cases were as follows: uncertainty in identifying animal abuse (29%), concern about breaching the clients’ confidentiality (23.8%), lack of knowledge about how to stop abuse and support victims (21.8%), and fear of losing income and clients (14%). The responses to open questions regarding barriers included the fear of the abusers’ retaliation (4%), lack of civil immunity laws to protect those who report abuse in good faith, and the shortage of time to intervene because of the overload at work. Only 8.9% of the participants had guidelines or helpful resources to aid in managing suspected animal abuse cases.

### 3.4. Predictors of the Veterinarians’ Intention to Intervene in Animal Abuse Cases

We assessed the results of three models for predicting the participants’ intention toward counseling and reporting on encountering animal abuse cases during their practice using a hierarchical multiple regression analysis (Table 6). To determine the effect of the predictors, the following factors were entered into model 1: sex, age, practice type, and “pro-animal” attitude. The belief in the “link” between animal abuse and domestic abuse or other types of crimes was added into the second model, and the belief in a moral and legal obligation to intervene in animal abuse cases was entered as a predictor into the third model.

Regarding counseling, a “pro-animal” attitude measured using the AAS-10 was the significant independent variable identified in model 1 (R^2^ = 0.025, F = 3.79, *p* < 0.01). In the second model, age and belief in the “link” between animal abuse and other types of crimes were found to be significantly related to the likelihood of a veterinarian counseling an animal owner (R^2^ = 0.095, F = 10.216, *p* < 0.001). The addition of the belief in duty factor into the third model resulted in an increase in R^2^ (ΔR^2^ = 0.138), and this explained the 23.2% of variance associated with the intention to counsel owners (F = 22.089, *p* < 0.001). In model 3, moral obligation was found to be the statistically significant predictor (β = 0.389, *p* < 0.001).

In the first model, a “pro-animal” attitude was found to be the most significant predictor (β = 0.297, *p* < 0.001) of the intention to report animal abuse with a percentage of explanation of 8.4% (F = 13.522, *p* < 0.001). In model 2 (R^2^ = 0.133, F = 14.934, *p* < 0.001), a “pro-animal” attitude (β = 0.236, *p* < 0.001) and the belief in the “link” between animal abuse and other human crimes (β = 0.267, *p* < 0.001) were found to have significant correlations with the participants’ intention to report. In the final model of the regression analysis, adding belief in duty subscales resulted in a positive R-squared change (ΔR^2^ = 0.106); therefore, the model explained the 23.9% (F = 22.877, *p* < 0.001) of variance of the dependent variable. Statistically, a “pro-animal” attitude (β = 0.168, *p* < 0.001), belief in the “link” to other types of crimes (β = 0.215, *p* < 0.001), and a sense of moral (β = 0.168, *p* < 0.001) and legal obligation (β = 0.238, *p* < 0.001) were significant predictors of the respondents’ intention to report animal abuse cases.

## 4. Discussion

As a professional who takes an oath to protect animal health and welfare, a veterinarian should have moral and ethical sensitivity, be familiar with legal liability, be equipped with good communication skills to connect with and understand clients, and pay attention to and continuously monitor issues related to an animal’s wellbeing. Additionally, developing self-efficacy in animal abuse reporting along with attending specialized training should be taken into consideration [33].

Over 80% of the veterinarians who participated in this study had witnessed suspected animal abuse in their practice, which is similar to the results (87%) observed in the United States [22] and is higher than those observed in New Zealand (63%) [24]. The relatively high incidence of animal abuse recognized by veterinary practitioners in South Korea may be due to a sharp increase in the pet population and growing publicity on the issue. However, inconsistencies in the animal abuse witness rates were found in the participant subgroups in our study. Practitioners from the groups consisting of female participants, small animal practitioners, and younger veterinarians reported on witnessing a larger number of animal abuse cases. Respondents with a stronger “pro-animal” attitude also reported on witnessing a larger number of animal abuse cases. These findings demonstrate that the veterinarians’ sensitivity and their threshold for recognizing NAIs as cruelty may differ among groups based on their attitude toward animals, professional experience, and animal welfare education levels.

Although the Korean Animal Protection Act (KAPA) was first enacted in 1991 and has prohibited inhumane treatment of animals, animals in South Korea have been exposed to the social ignorance and cruelty. A traditional “good to pet and eat” attitude still remains [34]. However, as the pet population increases [35], “pro-animal” attitude in public seems to be rapidly strengthening. A year-long campaign against dog meat led to the Supreme court judgement in 2018 that the killing dogs for meat is illegal. Changes in animal attitude have affected public expectations of veterinarians, their education and practice. Until the 1990s, the Korean veterinary curriculum did not include animal welfare or animal ethics. But, since 2000, animal behavior, animal welfare and veterinary ethics have been taught at veterinary schools in South Korea. Therefore, different age groups may have different professional expectations and sensitivity to animal abuse.

Counseling animal owners or having discussions with them in suspected animal abuse cases is an important strategy as an initial intervention that could prevent further abuse [12,22]. If the case is not manageable or is serious, it should be reported on without any delay in order to protect the victims and facilitate a more aggressive approach in the prosecution of individuals responsible for animal and domestic abuse [17,18]. In this study, on encountering a suspected animal abuse case, 74.6% of the respondents indicated that they would counsel the animal owners, and 48.6% indicated that they would report it. This implies that more than half of the participants were reluctant to report suspected cases to the relevant authorities. The majority of the respondents (68.1%) believed that the abused animal’s safety and wellbeing were still not guaranteed after reporting the case to the responsible authorities. The KAPA was amended over twenty times since 1991; however, the weak enforcement of the KAPA is still criticized because of the increase in the prevalence and intensity of animal abuse over time. Therefore, collaboration and communication between the regional government that is responsible for preventing animals abuse and the veterinary society are urgently required.

The concern regarding the veterinarian–client relationship was another factor that was frequently considered as a barrier by the participants in our study. Further, it was found that veterinary practitioners experienced anxiety with regard to concerns that they might lose their income, harm their reputation, or face retaliation and legal repercussions, which was in agreement with the findings of previous studies [22,25]. However, veterinarians have a responsibility toward both their patient and client based on the sound veterinarian–patient animal–client relationship (VPCR) guideline. Breaching client confidentiality can be considered as the lesser of two evils when taking into account the consequences of not reporting abuse [1]. Strengthening and improving the ethical decision-making skills that are imparted during veterinary education can help veterinarians in maintaining effective, balanced, and extended VPCRs.

Similar to the findings of a previous study [25], our findings revealed that participants lacked self-efficiency in managing animal abuse cases. They were not sure about how to identify animal abuse and were not confident about stopping the abuse and supporting the victim. Therefore, primary and continuing educational efforts or appropriate information resources on effective intervention in animal abuse cases are strongly required in order to assist veterinarians in dealing with such cases [23]. As part of their education programs, veterinarians should be provided with clear instructions on the following five steps that should generally be adopted for them to be able to professionally respond in a suspected animal abuse situation: building awareness, resolving contentious ethical dilemmas, providing guarantees of legal protection, expertise in identification of abuse indicators, and developing standardized protocols on responses that can aid in overcoming normative barriers [17,19].

Based on the results of the regression models, we found that moral obligation was the only significant predictor for providing counseling as an advisory and educational intervention (β = 0.389, *p* < 0.001). Meanwhile, a “pro-animal” attitude, moral and legal obligation, and the belief in the “link” between animal abuse and other types of crimes involving humans were found to be significantly related to the participants’ intention toward reporting the abuse as a legal intervention. This finding is partially along the same lines of those of previous studies that indicate that the belief in the “link” between animal and human abuse has positively influenced the veterinarians’ intention toward reporting suspected animal abuse cases [21,24]. A “pro-animal” attitude based on a concern for animals and the veterinarians’ moral and legal obligation toward animal patients can be reinforced via educational programs on animal welfare and veterinary professionalism. Furthermore, veterinarians should receive training in veterinary forensic medicine in order to be able to identify and record any injury, illness, or abnormality, and to interpret these findings in a way that could be admitted to a court of law for the officials to be able to understand the causes and implication of any changes. As an essential element associated with law enforcement and understanding a victim’s body language and unspoken signs indicating suffering or pain, these forensic educational training programs can facilitate the veterinarians’ active intervention in animal abuse cases [36,37,38].

The current study provided practical suggestions to strengthen veterinary practitioners’ role in animal abuse prevention, but there are some limitations. Although the participants’ intention based on their internalized moral belief has been acknowledged to develop a given behavior, the findings of this study may be limited by the intention–behavior gap [39]. Furthermore, animal abuse prevention involves collaboration among related professionals with citizens. Measuring trust in this collaboration and its effectiveness were beyond the scope of study, and therefore need to be further investigated.

## 5. Conclusions

This is the first study to investigate the intention of South Korean veterinary practitioners toward intervening in animal abuse cases, and our results demonstrated that individual characteristics (sex, age, practice type, “pro-animal” attitude) have an effect on their sensitivity in recognizing animal abuse. Their intention to intervene in animal abuse cases can be reinforced by strengthening their “pro-animal” attitude and their belief in the “link” between animal abuse and human abuse and in their moral and legal obligations toward animals. Our results prove that there is an urgent requirement to develop educational programs that focus on legal liability, moral responsibility, and veterinary forensic medicine in order to strengthen the veterinarians’ competency and self-efficacy in identifying NAIs caused by abuse. Moreover, appropriate protocols and protective legislative guidelines are required in order to secure an animal’s and a reporter’s safety after a suspected case is reported.

## Figures and Tables

**Table 1 animals-10-00802-t001:** Respondents’ individual characteristics and responses to witnessing animal abuse.

Individual Characteristics	N (%)	Pro-animal Attitude M (SD)	Witnessed Animal Abuse	
			Yes	No
**Gender**		t = −10.08, *p* < 0.001	χ^2^ = 7.313, *p* < 0.01	
Females	137 (23.1%)	34.6 (4.6)	128 (93.4%)	9 (6.6%)
Males	456 (76.9%)	30.4 (4.3)	385 (84.4%)	71 (15.6%)
**Age Group**		F = 6.675, *p* < 0.001	χ^2^ = 23.537, *p* < 0.001	
Under 30 y	46 (7.8%)	33.4 (3.6) ^a^	43 (93.5%) ^a^	3 (6.5%)
30–39 y	173 (29.2%)	32.1 (4.9) ^ab^	157 (90.8%) ^a^	16 (9.2%)
40–49 y	222 (37.4%)	31.2 (5.1) ^bc^	196 (88.3%) ^a^	26 (11.7%)
50–59 y	106 (17.9%)	30.4 (4.4) ^c^	86 (81.1%) ^ab^	20 (18.9%)
60–69 y	34 (5.7%)	29.1 (3.6) ^c^	24 (70.6%) ^b^	10 (29.4%)
Over 70 y	12 (2%)	28.2 (2.6) ^c^	7 (58.3%) ^b^	5 (41.7%)
**Practice type**		t = −4.92, *p* < 0.001	χ^2^ = 20.937, *p* < 0.001	
Large animal	81 (13.7%)	29 (4)	57 (70.4%)	24 (29.6%)
Small animal	512 (86.3%)	31.7 (4.7)	456 (89.1%)	56 (10.9%)
Total	593 (100%)	31.3 (4.7)	513 (86.5%)	80 (13.5%)

*Note*: ^abc^ Means indicated with the same letters are not significantly different at *p* < 0.05 in a Tukey HSD post hoc comparison. M: mean, SD: standard deviation, HSD: honestly significant difference.

**Table 2 animals-10-00802-t002:** Frequency of witnessing animal abuse cases per year (*n* = 593).

Frequency	None	Less than Once	2–3 Times	4–11 Times	More than 12 Times
*n*	80 (13.5%)	177 (29.8%)	177 (29.8%)	93 (15.7%)	66 (11.1%)

**Table 3 animals-10-00802-t003:** Belief in the “link” and obligation to intervene in animal abuse cases.

Belief	Strongly Agree	Agree	Neutral	Disagree	Strongly Disagree
**Belief—abuse**	197 (33.2%)	240 (40.5%)	92 (15.5%)	51 (8.6%)	13 (2.2%)
**Belief—crime**	203 (34.2%)	245 (41.3%)	94 (15.9%)	42 (7.1%)	9 (1.5%)
**Moral obligation**	96 (16.2%)	330 (55.6%)	118 (19.9%)	38 (6.4%)	11 (1.9%)
**Legal obligation**	36 (6.1%)	135 (22.8%)	167 (28.2%)	174 (29.3%)	81 (13.7%)

*Note*. Belief—abuse = belief in the “link” between animal abuse and domestic abuse, belief—crime = belief in the “link” between animal abuse and other types of human crimes.

**Table 4 animals-10-00802-t004:** Respondents’ intention toward being involved in suspected animal abuse cases.

Intervention	Definitely Do	Probably Do	Neutral	Probably Do Not	Definitely Do Not
**Counseling**	208 (35.1%)	234 (39.5%)	96 (16.2%)	48 (8.1%)	7 (1.2%)
**Reporting**	89 (15%)	199 (33.6%)	156 (26.3%)	120 (20.2%)	29 (4.9%)

**Table 5 animals-10-00802-t005:** Barriers associated with intervening in suspected animal abuse cases.

Barriers	N (%)
Difficulty in assuring the animal’s safety and welfare after reporting a suspected animal abuse case	404 (68.1%)
Uncertainty in identifying animal abuse	172 (29%)
Concern about breaching clients’ confidentiality	141 (23.8%)
Lack of knowledge about how to stop abuse and support victim	129 (21.8%)
Fear of losing income and clients	83 (14%)
Others (open answers)	58 (9.8%)

**Table 6 animals-10-00802-t006:** Hierarchical regression analysis predicting the veterinary practitioners’ intention to intervene in animal abuse cases.

Categories	Variables	Model 1 (Personal)			Model 2 (Belief-Link)			Model 3 (Belief-Duty)		
		B	SE	β	B	SE	β	B	SE	β
**Intention** **to counsel**	Sex	0.055	0.105	0.024	−0.001	0.101	0.000	0.063	0.094	0.028
	Age	0.071	0.040	0.080	0.084	0.038	0.094 ^*^	0.051	0.036	0.057
	Practice type	0.037	0.121	0.013	−0.031	0.118	−0.011	0.046	0.109	0.016
	AAS-10	0.030	0.009	0.147 ^**^	0.014	0.009	0.068	−0.001	0.009	−0.003
	Belief—abuse				0.097	0.065	0.101	0.005	0.061	0.006
	Belief—crime				0.197	0.069	0.195 ^**^	0.122	0.064	0.121
	Moral obligation							0.440	0.051	0.389 ^***^
	Legal obligation							0.040	0.037	0.046
**Variance**	R2	0.025			0.095			0.232		
	Δ22				0.07			0.138		
	F	3.790 ^**^			10.216 ^***^			22.089 ^***^		
**Intention** **to report**	Sex	0.023	0.116	0.009	−0.023	0.113	−0.009	0.052	0.106	0.020
	Age	0.046	0.044	0.045	0.057	0.043	0.056	0.018	0.040	0.017
	Practice type	−0.076	0.134	−0.024	−0.144	0.131	−0.045	−0.063	0.124	−0.020
	AAS-10	0.070	0.010	0.297 ^***^	0.055	0.010	0.236 ^***^	0.040	0.010	0.168 ^***^
	Belief—abuse				−0.045	0.073	−0.041	−0.117	0.069	−0.107
	Belief—crime				0.307	0.077	0.267 ^***^	0.248	0.073	0.215 ^***^
	Moral obligation							0.216	0.058	0.168 ^***^
	Legal obligation							0.234	0.042	0.238 ^***^
**Variance**	R2	0.084			0.133			0.239		
	ΔR2				0.048			0.106		
	F	13.522 ^***^			14.934 ^***^			22.877 ^***^		

Note. Belief—abuse = belief in the “link” between animal abuse and domestic abuse, belief—crime = belief in the “link” between animal abuse and other types of human crimes, AAS-10 = animal attitude scale-10, SE = standard error. **^*^**
*p* < 0.05; ^**^
*p* < 0.01; ^***^
*p* < 0.001.

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
