# Peer review of "An Analysis of Veterinary Practitioners’ Intention to Intervene in Animal Abuse Cases in South Korea"

_animals, 2020, doi:10.3390/ani10050802_

Round 1
Reviewer 1 Report
Dear Author,
These are my questions/ comments:
136-138:
We received a total of 137 1,818 responses (26.7%). To analyze animal abuse-related items, we used 593 completed questionnaires from small and large animal practitioners (32.6% of the responses).
Question:
How were the 593 questionnaires selected? Were those the only questionaires that were completed? Then it needs to be stated. If that was not the case, how were they selected?
149-153:
The participants (n = 593, 76.9% male and 23.1% female) whose responses to our survey were analyzed represented approximately 8.4% of the total veterinary practitioners in Korea (n = 7,099 in January of 2019 per the KVMA statistics). The majority of the participants were in their 40s (37.4%), followed by in their 30s (29.2%). Approximately 86% of the participants were small animal practitioners (Table 1).
Question: Are the participants or chosen questionnaire responders ,also representative for the total Korean´ vets regarding sex, age and ratio between small and large animal practitioners?
192-193
Note. Belief-abuse = belief in the “link” between animal abuse and domestic abuse, Belief-crime = belief in the “link” between animal abuse and other types of human crimes
219-220
Note. Belief-abuse = belief in the “link” between animal abuse and domestic (child / women) abuse, Belief-crime = belief in the “link” between animal abuse and other forms of human crimes
Comment: 192-193 and 219-220: Notes/ explanation should be identical
Author Response
We appreciate the time and effort that you have dedicated to providing valuable feedback on my manuscript. We are grateful to you for your insightful comments on this paper. We have been able to incorporate changes to reflect most of your suggestions. We have highlighted the changes within the manuscript.
Here is a point-by-point response to your comments and concerns.
Line 136-138:
We received a total of 137 1,818 responses (26.7%). To analyze animal abuse-related items, we used 593 completed questionnaires from small and large animal practitioners (32.6% of the responses).
Question: How were the 593 questionnaires selected? Were those the only questionnaires that were completed? Then it needs to be stated. If that was not the case, how were they selected?
Answer: The whole questionnaire intended to investigate ethical issues in veterinary medicine in South Korea, regardless specialty, as we mentioned in the text (line 132-134). The questions about animal abuse were for only small and large animal practitioners. We restated the sentences (lines 135 to 139) and changed the title to clarify the participants (from veterinarians to veterinary practitioners).
- [Line 135-139] We received a total of 1,818 responses (26.7% response rate) to the survey on ethical issues in Korea, in which a section of animal abuse questions exclusively for small and large animal practitioners were included. In this survey we selected the 593 responses from small and large animal practitioners (32.6% of the total responses) and analyzed the animal abuse questions.
Line 149-153:
The participants (n = 593, 76.9% male and 23.1% female) whose responses to our survey were analyzed represented approximately 8.4% of the total veterinary practitioners in Korea (n = 7,099 in January of 2019 per the KVMA statistics). The majority of the participants were in their 40s (37.4%), followed by in their 30s (29.2%). Approximately 86% of the participants were small animal practitioners (Table 1).
Question: Are the participants or chosen questionnaire responders ,also representative for the total Korean´ vets regarding sex, age and ratio between small and large animal practitioners?
Answer: We asked for updated population data of Korean veterinary practitioners from KVMA and reconfirmed that our chosen questionnaire responders are representative for total vets regarding sex, age and ratio between small and large animal practitioners. We have, accordingly, revised the sentences in the line 149 to 154.
- [Line 149-154] The chosen questionnaire responders (n = 593, 76.9% male and 23.1% female) represented approximately 8.5% of the total veterinary practitioners in Korea (n = 6,972 in January of 2020 per the KVMA statistics). The majority of the responders were in their 40s (37.4%), followed by in their 30s (29.2%). Approximately 86% of the participants were small animal practitioners (Table 1). Participants represented the population of veterinary practitioners in Korea, in terms of gender (male 79.5%, female 20.5%), practice type (small animal 86.9%, large animal 13.1%), and age (under the 50s 68.8%).
Line 192-193:
Note. Belief-abuse = belief in the “link” between animal abuse and domestic abuse, Belief-crime = belief in the “link” between animal abuse and other types of human crimes
Line 219-220:
Note. Belief-abuse = belief in the “link” between animal abuse and domestic (child / women) abuse, Belief-crime = belief in the “link” between animal abuse and other forms of human crimes
Comment: 191-192 and 217-218: Notes/ explanation should be identical
Answer: We have changed the Notes.
- [Line 188-189] Note. Belief-abuse = belief in the “link” between animal abuse and domestic abuse, Belief-crime = belief in the “link” between animal abuse and other types of human crimes.
- [Line 214-215] Note. Belief-abuse = belief in the “link” between animal abuse and domestic abuse, Belief-crime = belief in the “link” between animal abuse and other types of human crimes, …
Reviewer 2 Report
Below are my comments in review of "An analysis of the veterinarians' intention to intervene in animal abuse cases in Korea."
Overall, I find it interesting to read about veterinarian obligations and intention to report animal abuse in South Korea given that, less than a decade ago, dogs and cats were still in a space between food and companion. This alone makes the intellectual merit of this paper important, as I think it is a valuable contribution to the changing landscape of animals in many parts of Asia.
Grammar
The grammar and spelling of the paper do need a close editor's eye, but it is generally done well. For example, in the title, remove "the" and leave "analysis of veterinarians' intention..." Likewise, the first sentence in the Introduction is a significant run on, with many ideas covering 4-5 lines of text. Try not to challenge your reader's attention span so soon.
Specific Comments
As stated above, I think this article has a valuable contribution to make. This is especially so given that the article emphasizes South Korean (and for a global audience, I would address the nation as South Korea) veterinarians. Because of this, a brief acknowledgement of the history of animals in South Korea is appropriate. This will further validate the importance of the findings. Once the reader has a firmer grasp of the changing roles of pets and other animals in South Korea, the significance of who chooses to report/intervene and why becomes even more salient.
Introduction
1. Second paragraph: Reorganize your thoughts for this paragraph around the idea of mandatory vs. nonmandatory reporting. It's a bit of whiplash for the reader to open this paragraph with statements about battered child syndrome. Rather, if you open by stating that there is a wide variation in mandatory vs. nonmandatory legislation across nations, you can use the battered child syndrome as the example it was intended to be. From there, you can move through the rest of the ideas regarding what countries require reporting and where South Korea fits into that landscape.
2. Line 90: remove "on." "Therefore, previous studies have emphasized making..."
Materials and Methods
1. Give consideration to providing the full survey as an appendix. There is debate on this, but I lean toward the idea that it helps your reader visualize the questions and how they are related.
2. What happened to your sample? You received 1,818 responses, but then chose only to use 593 in your analysis. Please detail what happened to the 1225 missing responses? Were they incomplete? Did you have other exclusion criteria that were not met? This needs a clear explanation, otherwise, your reader spends the rest of the paper wondering if and how those additional responses would shift the analysis.
Results
1. Confirm that there is no statistical noise created by the removal of responses. Also, consider that your ratio of men to women may create statistical errors. 76.9% to 23.1% is getting close to violating central theorem limits.
2. Line 203: Add "be." "...found to be the perceived..."
3. Did you ask about why/how the vets thought they would lose income?
4. Have you looked into the training veterinarians receive in South Korea? Again, given the history the nation has with animals, perhaps the older veterinarians simply have a different sense of the purpose of animals or expectations of "senscience?"
Discussion
Here is a great place to discuss the history of animals' "purpose" in South Korea and how it may be influencing veterinarians' perceptions of "abuse." For example, Podberseck (2009) talked about informants' conflict between loving their pet but wanting others in society to have the autonomy to still eat meat. Since then, legislation has made it illegal to raise dog specifically for food. This may be extremely relevant to how your population is responding to the survey (why there is an age or sex difference). It would also flesh out your paper from research report to academic article.
I hope these recommendations, questions, and comments serve you well in revising the paper. Good luck!
Author Response
We appreciate the time and effort that you have dedicated to providing valuable feedback on my manuscript. We are grateful to you for your insightful comments on this paper. We have been able to incorporate changes to reflect most of your suggestions. We have highlighted the changes within the manuscript.
Here is a point-by-point response to your comments and concerns.
Comments and Suggestions for Authors
Below are my comments in review of "An analysis of the veterinarians' intention to intervene in animal abuse cases in Korea."
Overall, I find it interesting to read about veterinarian obligations and intention to report animal abuse in South Korea given that, less than a decade ago, dogs and cats were still in a space between food and companion. This alone makes the intellectual merit of this paper important, as I think it is a valuable contribution to the changing landscape of animals in many parts of Asia.
Grammar
The grammar and spelling of the paper do need a close editor's eye, but it is generally done well. For example, in the title, remove "the" and leave "analysis of veterinarians' intention..." Likewise, the first sentence in the Introduction is a significant run on, with many ideas covering 4-5 lines of text. Try not to challenge your reader's attention span so soon.
Answer: We have, accordingly, modified the title and the first sentence of the introduction.
- Title: An analysis of veterinary practitioners’ intention to intervene in animal abuse cases in South Korea
- [Line 43-45] Animal abuse is not only an animal welfare issue, but it is also a critical concern to the wellbeing of humans. It is considered as a sign of antisocial behavior trends [1] based on the potential “link” between animal abuse and child abuse or other forms of domestic violence [2-6].
Specific Comments
As stated above, I think this article has a valuable contribution to make. This is especially so given that the article emphasizes South Korean (and for a global audience, I would address the nation as South Korea) veterinarians. Because of this, a brief acknowledgement of the history of animals in South Korea is appropriate. This will further validate the importance of the findings. Once the reader has a firmer grasp of the changing roles of pets and other animals in South Korea, the significance of who chooses to report/intervene and why becomes even more salient.
Answer:
- We changed the nation name Korea in our manuscript to South Korea.
- Definitely there is a generation gap in perceiving animals in society, because attitude toward animals in our society is rapidly changing. We added a paragraph on the cultural shift in South Korea in the discussion section.
Introduction
- Second paragraph: Reorganize your thoughts for this paragraph around the idea of mandatory vs. nonmandatory reporting. It's a bit of whiplash for the reader to open this paragraph with statements about battered child syndrome. Rather, if you open by stating that there is a wide variation in mandatory vs. nonmandatory legislation across nations, you can use the battered child syndrome as the example it was intended to be. From there, you can move through the rest of the ideas regarding what countries require reporting and where South Korea fits into that landscape.
Answer: As you commented we reorganized the paragraph to start with the variation in registration across nations (Line 45-68).
- Line 90: remove "on." "Therefore, previous studies have emphasized making..."
Answer: We have, accordingly, corrected this grammar error.
- [Line 89-90] Therefore, previous studies have emphasized making animal abuse reporting mandatory by law and on ensuring legal immunity to support the veterinarians who report on such cases [7,20,29].
Materials and Methods
- Give consideration to providing the full survey as an appendix. There is debate on this, but I lean toward the idea that it helps your reader visualize the questions and how they are related.
Answer: Thank you for this suggestion. We uploaded the full survey as an appendix with the revised manuscript. But we do not think it is necessary to provide the full survey, because the questions were simple and the most of the answers were presented in the result.
- What happened to your sample? You received 1,818 responses, but then chose only to use 593 in your analysis. Please detail what happened to the 1225 missing responses? Were they incomplete? Did you have other exclusion criteria that were not met? This needs a clear explanation, otherwise, your reader spends the rest of the paper wondering if and how those additional responses would shift the analysis.
Answer: The whole questionnaire intended to investigate ethical issues in veterinary medicine in South Korea, regardless specialty, as we mentioned in the line 132-134. The questions about animal abuse were for only small and large animal practitioners. We restated the sentences in the lines 135 to 139 and changed the title to clarify the participants (from veterinarians to veterinary practitioners).
- [Line 135-139] We received a total of 1,818 responses (26.7% response rate) to the survey on ethical issues in Korea, in which a section of animal abuse questions exclusively for small and large animal practitioners were included. In this survey selected the 593 responses from small and large animal practitioners (32.6% of the total responses) and analyzed the animal abuse questions.
Results
- Confirm that there is no statistical noise created by the removal of responses. Also, consider that your ratio of men to women may create statistical errors. 76.9% to 23.1% is getting close to violating central theorem limits.
Answer: You have raised an important point here. Due to the ratio of men and women, it seems to close to violating central theorem limits, but our sample size (female n=137) is large enough for analysis without statistical errors. We referenced the following article.
Kwak, S. G., & Kim, J. H. (2017). Central limit theorem: the cornerstone of modern statistics. Korean journal of anesthesiology, 70(2), 144-156. (https://doi.org/10.4097/kjae.2017.70.2.144)
- Line 203: Add "be." "...found to be the perceived..."
Answer: We have, accordingly, corrected this grammar error.
- [Line 198-200 ] The most significant barrier associated with intervening in suspected animal abuse cases was found to be the perceived difficulty in assuring the animal’s safety and welfare after reporting a suspected animal abuse case.
- Did you ask about why/how the vets thought they would lose income?
Answer: We didn’t ask about the details about the fear of losing income in this survey. But we could anticipate the reasons why they worry about that matters from the veterinarians’ open answers about barriers in our survey; it may be related with retaliation, disrepute, damage of their relationship with clients, and spending time and efforts to intervene animal abuse in their work time.
- Have you looked into the training veterinarians receive in South Korea? Again, given the history the nation has with animals, perhaps the older veterinarians simply have a different sense of the purpose of animals or expectations of "senscience?"
Answer: We added a paragraph on the cultural shift in South Korea in the discussion section.
Discussion
Here is a great place to discuss the history of animals' "purpose" in South Korea and how it may be influencing veterinarians' perceptions of "abuse." For example, Podberseck (2009) talked about informants' conflict between loving their pet but wanting others in society to have the autonomy to still eat meat. Since then, legislation has made it illegal to raise dog specifically for food. This may be extremely relevant to how your population is responding to the survey (why there is an age or sex difference). It would also flesh out your paper from research report to academic article.
Answer:
- [Line 262-271] Although the Korean Animal Protection Act (KAPA) was first enacted in 1991 and has prohibited inhumane treatment of animals, animals in South Korea have been exposed to the social ignorance and cruelty. Traditional “good to pet and eat” attitude still remains [38]. However, as the pet population increases [39], pro-animal attitude in public seems to be rapidly strengthening. A year-long campaign against dog meat led to the Supreme court judgement in 2018 that the killing dogs for meat is illegal. Changes in animal attitude have affected public expectations of veterinarians, their education and practice. Until the 1990s veterinary curriculum did not include animal welfare or animal ethics. But, since 2000, animal behavior, animal welfare and veterinary ethics have been taught at veterinary schools in South Korea. Therefore, different age groups may have different professional expectations and sensitivity to animal abuse.
I hope these recommendations, questions, and comments serve you well in revising the paper. Good luck!
- Thank you so much!
Reviewer 3 Report
None.
Author Response
We appreciate the time and effort that you have dedicated to providing valuable feedback on my manuscript.
Reviewer 4 Report
Excellent study, timely, international significance!
Author Response

(The authors gave the same response as above.)
